Clinicopathological value of long non-coding RNA profiles in gastrointestinal stromal tumor

http://orcid.org/0000-0001-7279-3829 Zhao Yan 1
Liu Xinxin 2 gorilla1999@hotmail.com
Xiao Keshuai 3
Wang Liwen 2
http://orcid.org/0000-0003-1715-7787 Li Yuping 2
Kan Mingyun 2
Jiang Zhiwei 2 surgery34@163.com
1 Clinical Medical College of Yangzhou University, Northern Jiangsu People’s Hospital , Yangzhou, Jiangsu Province , China
2 Department of Gastrointestinal Surgery, Jiangsu Province Hospital of Traditional Chinese Medicine , Nanjing, Jiangsu Province , China
3 Department of General Surgery, Yangzhou Hongquan Hospital , Yangzhou, Jiangsu Province , China
Zhao Min
Electronic publication date: 2021 Sep 3
Publication date: 2021
Volume: 9
Electronic Location ID: e11946
Received 2020 Dec 23; Accepted 2021 Jul 20
Copyright: © 2021 Zhao et al.
Copyright year: 2021
Copyright holder: Zhao et al.
License: This is an open access article distributed under the terms of the Creative Commons Attribution License, which permits unrestricted use, distribution, reproduction and adaptation in any medium and for any purpose provided that it is properly attributed. For attribution, the original author(s), title, publication source (PeerJ) and either DOI or URL of the article must be cited.
License URL: https://creativecommons.org/licenses/by/4.0/

Keywords: GIST, LncRNA, NMF consensus clustering, SEER, LRRC75A-AS1

Funding: National Natural Science Funding of China 81300721 Science and Technology Development Funding of Yangzhou City YZ2014204 Social Development Fund of Jiangsu Province BE2015687 This study was supported by the grants from the National Natural Science Funding of China (No. 81300721), the Science and Technology Development Funding of Yangzhou City (No. YZ2014204), and the Social Development Fund of Jiangsu Province (No. BE2015687). The funders had no role in study design, data collection and analysis, decision to publish, or preparation of the manuscript.

==============================
Background

Long non-coding RNAs (lncRNAs) have been implicated in diagnosis and prognosis in various cancers. However, few lncRNA signatures have been established for prediction of gastrointestinal stromal tumors (GIST). We aimed to explore a lncRNA signature profile that associated with clinical relevance by mining data from Gene Expression Ominus (GEO) and Surveillance, Epidemiology, and End Results (SEER) Program.

Methods

Using a lncRNA-mining approach, we performed non-negative matrix factorization (NMF) consensus algorithm in Gastrointestinal stromal tumors (GISTs) cohorts (61 patients from GSE8167 and GSE17743) to cluster LncRNA expression profiles. Comparative markers selection, and Gene Set Enrichment Analysis (GSEA) algorithm were performed between distinct molecular subtypes of GIST. The survival rate of GIST patients from SEER stratified by gender were compared by Kaplan–Meier method and log-rank analysis. lncRNA-mRNA co-expression analysis was performed by Pearson correlation coefficients (PCC) using R package LINC. Somatic copy number alterations of GIST patients (GSE40966) were analyzed via web server GenePattern GISTIC2 algorithm.

Results

A total of four lncRNA molecular subtypes of GIST were identified with distinct biological pathways and clinical characteristics. LncRNA expression profiles well clustered the GIST samples into small size (<5 mm) and large size tumors (>5 mm), which is a fundamental index for GIST malignancy diagnosis. Several lncRNAs with abundant expression (LRRC75A-AS1, HYMAI, NEAT1, XIST and FTX) were closely associated with tumor size, which may suggest to be biomarkers for the GIST malignancy. Particularly, LRRC75A-AS1 was positively associated with tumor diameters and suggested an oncogene in GIST. Co-expression analysis suggested that chromosome region 17p11.2–p12 may contribute to the oncogenic process in malignant GIST. Interestingly, the gender had a strong influence on clustering by lncRNA expression profile. Data from the Surveillance, Epidemiology, and End Results (SEER) Program were further explored and 7983 patients who were diagnosed with GISTs from 1973 to 2014 were enrolled for analysis. The results also showed the favorable prognosis for female patients. The survival rate between male and female with GIST was statistically significant (P < 0.0001). Gene set enrichment analysis (GSEA) indicated distinct pathways between female and male, and malignant GIST was associated with several cancer metabolism and cell cycle associated pathways.

Conclusions

This lncRNAs-based classification for GISTs may provide a molecular classification applicable to individual GIST that has implications to influence lncRNA markers selection and prediction of tumor progression.

Introduction

Gastrointestinal stromal tumors (GISTs) constitute the most common form of subepithelial tumor of the gastrointestinal tract (Xu et al., 2014; Yan et al., 2015). Approximately 60% to 70% of GISTs arise in the stomach, 20% to 30% in the small intestine, and 5% in the colon and rectum (Corless, Fletcher & Heinrich, 2004; Miettinen & Lasota, 2001). On the basis of similarities in immunohistochemical and ultrastructural features, it is considered that GISTs arise from interstitial cells of Cajal (ICC) or their precursor cells (Kindblom et al., 1998). More than 80% of GISTs have gain of function mutations of the KIT proto-oncogene that encodes the c-Kit (CD117) (Hirota et al., 1998), and the PDGFRA gene that encodes platelet-derived growth factor receptor (PDGFRA) tyrosine kinase (Hirota et al., 2003). Nevertheless, GISTs present extremely heterogeneous clinical prognosis. Currently, the criteria for risk estimation depend largely on clinicopathologic factors, such as tumor site, size, cell type, degree of necrosis, mitotic rate, Ki-67 immunoreactivity as well as their combinations (DeMatteo et al., 2000; Miettinen et al., 2002).

Increasing evidence suggested that the aberrant expression of long non-coding RNAs (lncRNAs) have been associated with cancer initiation and progression (Mitra, Mitra & Triche, 2012; Yoon, Kim & Gorospe, 2015), and some of them have been implicated in diagnosis and prognosis (Qi & Du, 2013). Moreover, lncRNAs emerge as strong cancer biomarkers as it has higher specificity than protein-coding mRNAs (Hessels et al., 2003; Prensner & Chinnaiyan, 2011), and stably detectable in the blood (Lin et al., 2007) and urine (Hessels et al., 2003; Tinzl et al., 2004) by conventional RT-PCR methods. In GIST, studies have shown that lncRNA HOTAIR could be one of the best candidates as potential prognostic biomarkers (Lee et al., 2016; Niinuma et al., 2012). FFPE tissue specimens data from 40 surgically resected and metastatic GIST patients suggest a potential biomarker and prognostic value of both H19 and MALAT1 lncRNAs for the clinical selection of the best candidate to first-line treatment with imatinib (Badalamenti et al., 2019). LncRNA AOC4P in 79 GIST patients’ tissues in high risk GIST tissues was higher than that in low/medium-risk GIST tissues and normal tissues (Hu et al., 2018). In vitro cell culture study showed lncRNA AOC4P silencing can reduce the proliferative ability, decrease the migration and invasion activity, and induce cell apoptosis (Hu et al., 2018). Studies by using chip technology also identified a panel of dysregulated lncRNAs that may serve as potential biomarkers or drug targets for GISTs, particularly secondary imatinib-resistant GISTs (Yan et al., 2019). Therefore, searching a lncRNA signature might suggest a potential biomarker and prognostic value for GIST.

LncRNA profiling analysis could be realized by mining previously published gene expression microarray due to a large group of lncRNA-specific probes present on the commonly used microarray platforms. Multiple studies have discovered new lncRNA biomarkers and identified therapeutic lncRNA targets using mining approach of re-annotating microarray (Zhang et al., 2012; Zhu et al., 2016). In this study, we analyzed a well-characterized cohort of GIST microarrays from the Gene Expression Omnibus (GEO) in order to clarify the gene expression or lncRNA alterations associated with clinicopathological value. The Surveillance, Epidemiology, and End Results (SEER) Program of the US National Cancer Institute (NCI) was also used to identify GIST patients’ overall survival and investigate molecular features that might be applicable to the prediction of outcome.

Materials & Methods

GIST gene expression datasets preparation

Microarray data of GSE20710_RAW (Astolfi et al., 2010), GSE56670_RAW (Killian et al., 2014), GSE8167_RAW (Yamaguchi et al., 2008) and GSE17743_RAW (Ostrowski et al., 2009) were directly downloaded from Gene Expression Omnibus databases. These datasets corresponded to all available public datasets fulfilling the following criteria: available gene expression data obtained using the same chip platform (U133 Plus 2.0 chips, Affymetrix, Santa Clara, CA, USA) with raw data CEL files, and some associated clinicopathological data were available. SDH-deficient gastrointestinal stromal tumors (Killian et al., 2014) were excluded, because of the small percentage of GIST and totally different gene-expression style (Janeway et al., 2011; Miettinen et al., 2011). Gene expression’s background correction, normalization and summarizing were performed using Robust Multichip Average (RMA) (Irizarry et al., 2003b) by TCA software (Affymetrix Co., Santa Clara, CA, USA) (Irizarry et al., 2003a).

Microarray data processing and lncRNA profile mining

The approach of lncRNA profile mining mainly referred to Hu et al. (2014), Zhu et al. (2016) and Zhang et al. (2012). Briefly, the Affymetrix HG-U133 Plus 2.0 probe set IDs were mapped to the NetAffx Annotation Files and gene biotypes were further validated by BioMart databases (biomaRt R package). Based on the Refseq transcript ID and/or Ensembl gene ID, we only retained non-coding protein genes and further filtered them by eliminating pseudogenes including microRNAs, rRNAs and other short RNAs such as snoRNAs, snRNAs and tRNAs. Finally, 2,448 annotated lncRNA transcripts with corresponding Affymetrix probe IDs were generated. Some lncRNA signatures were well confirmed by previous study (Hu et al., 2014; Zhu et al., 2016).

LncRNA markers selection and expression style between subtypes

The subtypes of GIST were identified in row median centered microarray datasets using the non-negative matrix factorization (NMF) algorithm (Brunet et al., 2004). We used the Comparative Marker Selection algorithm (Gould et al., 2006) by GenePattern (https://genepattern.broadinstitute.org/gp). One-vs.-all comparisons were performed to identify differentially expressed genes that can discriminate between distinct lncRNA clustering subtypes. Strict criteria were used to select the lncRNA markers, including fold change of gene expression > 2, FDR (BH) < 0.05, Feature-Specific P-value < 0.05.

Identification of biological pathways distinguishing each subtype

We performed GSEA through the JAVA program (http://www.broadinstitute.org/gsea) by using MSigDB C2 CP: Canonical pathways gene set collection (1,320 gene sets available), and visualized the GSEA outputs in Cytoscape (version 3.4). Then we used the Enrichment Map software to identify the biological processes distinguishing one subtype from others. In order to simplify the network map, we chose a strict threshold of gene-set permutations with an FDR cutoff of 5% and P-value cutoff of 1% in Enrichment Map software.

SEER database exploration

SEER data was used to identify patients that were diagnosed as GISTs from 1973 to 2014 (Cronin, Ries & Edwards, 2014). The exclusion criterions included: age < 18, no evaluation of histological type, not the first diagnosed malignancy, an unknown cause of death or survival month. The primary outcomes of interest were overall survival (OS), which was calculated from the date of diagnosis to the date of attributed death.

To better explore gender influence on the survival rate, the propensity-score-matched analyses were performed with equal factors which may influence the prognosis. Pairing criteria included tumor sizes, grade, age, treatment. Survival function estimation was performed with the Kaplan–Meier method and the resulting curves compared with the log-rank test. All statistical analyses were computed using SPSS version 24 (IBM Corporation, Armonk, NY, USA).

lncRNA-mRNA co-expression network construction and gene annotation

R package LINC provided methods to compute co-expression networks of lincRNAs and protein-coding genes. Biological terms associated with the sets of protein-coding genes predicted the biological contexts of lincRNAs according to the ‘Guilty by Association’ approach (Gillis & Pavlidis, 2012). This R package LINC applied this idea on arbitrary expression matrices (Yu et al., 2015) and could reveal which functions, pathways or compartments were associated with the lncRNA co-expressed genes.

Somatic copy number alterations (CNA) frequency profiles

We downloaded CNA array data of patients via the NCBI Gene Expression Omnibus (accession number GSE20709), and analyzed the raw copy number data for each GIST sample provided by array comparative genomic hybridization (CGH) by using the Copy Number Inference Pipeline for SNP6 CEL files and GISTIC2 algorithm (Mermel et al., 2011) via web server GenePattern.

Results

Classification of GISTs into four subtypes based on extracted lncRNA profiles

Due to the limited chipset data available for GIST, only two datasets GSE8167_RAW (Yamaguchi et al., 2008) and GSE17743_RAW (Ostrowski et al., 2009) were adapted as main data source in this study. The clinicopathological, immunohistochemical, and genetic characteristics of the 61 cases of GIST used in the chipset analysis are summarized in Table S1.

To explore the overall gene expression pattern, we first performed unsupervised analysis of all 54,613 probe sets and with 20,647 collapsed genes. Consensus NMF clustering (Brunet et al., 2004) separated the 61 GISTs into three principal classes (max coef. = 0.9929, when k = 3). To eliminate probes that had little or no variation across samples, we then keep the 50% most varying probes to run the non-negative matrix factorization (NMF) algorithm again. Next, we repeated the same unsupervised analysis with extracted lncRNA probe sets. According to the cophenetic coefficient scores from the consensus NMF clustering (k = 2 to k = 6), we chose to cluster 61 GISTs into four subtypes (max coef. = 0.9932, when k = 4) (Figs. 1A–1F). The four clustering subtypes were designated as G1, G2, G3, and G4. Metagenes analysis and linkage tree of extracted lncRNAs ordered by k = 4 clustering subtypes were showed in Figs. 1G–1I. Thus, as previously reported (Brunet et al., 2004), NMF with consensus clustering and hierarchical dendrogram gave strong evidence for four classes split of the lncRNAs with a correspondingly high cophenetic coefficient.

Figure 1 Classification of GISTs into four subtypes based on extracted lncRNA profiles.

(A–F) Heatmap showing consensus NMF clustering analysis of discovery dataset GSE8167_RAW and GSE17743_RAW by extracted lncRNA probe sets. When K = 4, cophenetic coefficient was the maximum. (G–H) Metagenes analysis of extracted lncRNAs ordered by K = 4 clustering subtypes, with annotations of each subtype G1 to G4, and (I) height plot of linkage tree for the discovery dataset listed on the head of four subtypes heatmap, with annotations of each subtype G1 to G4.

LncRNAs clustering GIST subtypes were associated with tumor size and gender

Subtypes clustered by global gene expression (max coef. = 0.9929, k = 3) did not show any meaning to the clinicopathological value, as well as extracted most varying probes. However, clustering by extracted lncRNA profile showed distinct clinical relevance. Average tumor diameter in G1 or G2 subtype were small size (10.45 ± 7.90 mm) and were referred as benign or early stage the GIST. G3 or G4 subtype showed larger size (52.17 ± 30.89 mm), and were referred as malignant GIST tumor. Fisher’s exact test showed that there were significant differences between class G1 and class G2 in the frequency of gender, as well as between subtypes G1/G2 and subtypes G3/G4 in the frequency of tumor size. LncRNAs’ expression profile did not correlate with c-kit and PDGFRA mutation status, as well as age and tumor location (Table 1).

Table 1 Association analysis of clinical characteristics of GIST patients and 4 subtypes clustered by lncRNA profile.

Subtypes	Cases	Age	F:M	Location
intestinal:gastric	Mutation
KIT:WT:PDGRF	Diameter (mm)	Sig.
(p)	Diameter (mm)	Sig.
(p)	
G1	11	59.0 + 10.8	11:0	2:9	6:5:0	9.09 + 7.34	0.995a	10.45 + 7.90*	0.000c	
G2	20	63.6 + 12.4	1:19	7:13	16:4:0	11.2 + 8.28	
G3	11	–	3:8	0:11	6:1:4	48.73 + 25.16	0.921b	52.17 + 30.89*	
G4	19	–	4:15	0:19	10:2:7	54.16 + 34.27	
Notes:

GIST, gastrointestinal stromal tumors; F:M, Female: Male; KIT, c-kit; WT, wild type; PDGRF, platelet-derived growth factor receptor alpha.

a p values for diameter comparison between G1 and G2.

b p values for diameter comparison between G3 and G4.

c p values for diameter comparison between G1/2 and G3/4.

* p < 0.05.

Previous research showed that methylation pattern of GISTs arising from the stomach segregated apart from GISTs from the intestine (Huang et al., 2016). Unlike methylation pattern indicating that GIST tissue of origin has a strong imprint on the epigenetic profile of this disease (Huang et al., 2016), lncRNA expression profile does not have correlation to stomach or intestine origin. Instead, the gender has significant influence on clustering by lncRNA expression profile, especially in small size GIST. This may be due to that the same cajai cells originate from alimentary track no matter where the GISTs locate. The high expression abundance of gender-related lncRNA, such as XIST, FTX and TTY among the limited lncRNA-probes may also give much weight coefficient for this gender-oriented clustering.

Female GIST patients showed favorable survival time by SEER database

Female and male patient showed distinct lncRNA expression profile especially in early stage of GIST and rapidly lost the gender-oriented clustering during the tumor progress, which suggest gender may be prognosis relevant as well as tumor size. So, we further explored the SEER database to analyze the survival rate differences between male and female GIST patients.

A total of 7,983 eligible GIST patients were enrolled during the 41-year study period. The baseline characteristics of GIST patients stratified by gender were summarized in Table 2. The median survival time was 100 months (95% CI [94.16–105.84]) in male and 130 months (95% CI [121.38–138.62]) in female GIST patients, female patients had a better overall survival (OS) than male patients. (HR = 1.277, P < 0.0001, Fig. 2A).

Figure 2 Cumulative survival curves of GIST patients stratified by gender from SEER database.

(A) Before and (B) after propensity score-match (PSM), Cumulative overall survival analysis of male and female GIST patients (P < 0.0001).

Table 2 Equilibrium comparison of baseline characteristics of GIST patients before and after propensity score matching. stratified by gender (male vs. female).

	Before matching			After matching		
Characteristics	Male	Female	p		Male	Female	p	
	4,139 (51.8%)	3,844 (48.2%)			2,567 (50%)	2,567 (50%)		
Tumor site			0.054				1.0	
Stomach	2,192 (53.0%)	2,188 (56.9%)			1,467 (57.1%)	1,467 (57.1%)		
Small intestine	1,177 (28.4%)	977 (25.4%)			716 (27.9%)	716 (27.9%)		
Rectum	138 (3.3%)	85 (2.2%)			37 (1.4%)	37 (1.4%)		
Colon	112 (2.7%)	107 (2.8%)			40 (1.6%)	40 (1.6%)		
Others	520 (12.6%)	487 (12.7%)			307 (12.0%)	307 (12.0%)		
Age			<0.001				1.0	
≤60	1,958 (47.3%)	1,607 (41.8%)			1,259 (49.0%)	1,259 (49.0%)		
>60	2,181 (52.7%)	2,237 (58.2%)			1,308 (51.0%)	1,308 (51.0%)		
Therapy			<0.001				1.0	
Surgery	3,190 (77.1%)	3,086 (80.3%)			2,090 (81.4%)	2,090 (81.4%)		
No surgery	911 (22.0%)	731 (19.0%)			473 (18.4%)	473 (18.4%)		
Unknown	38 (0.9%)	27 (0.7%)			4 (0.2%)	4 (0.2%)		
Marital status			<0.001				1.0	
Married	2,751 (66.5%)	1,861 (48.4%)			1,675 (65.3%)	1,675 (65.3%)		
Widowed	189 (4.6%)	773 (20.1%)			157 (6.1%)	157 (6.1%)		
Single/unmarried	678 (16.4%)	636 (16.5%)			438 (17.1%)	438 (17.1%)		
Divorced/separated	318 (7.7%)	386 (10.0%)			199 (7.8%)	199 (7.8%)		
Unknown	203 (4.9%)	188 (4.9%)			98 (3.8%)	98 (3.8%)		
Race			0.62				1.0	
White	2,891 (69.8%)	2,578 (67.1%)			1,849 (72.0%)	1,849 (72.0%)		
Black	680 (16.4%)	729 (19.0%)			399 (15.5%)	399 (15.5%)		
Others	568 (13.7%)	537 (14.0%)			319 (12.4%)	319 (12.4%)		
Grade			0.37				1.0	
I/II	715 (17.3%)	765 (19.9%)			467 (18.2%)	467 (18.2%)		
III/IV	482 (11.6%)	387 (10.1%)			230 (9.0%)	230 (9.0%)		
Unknown	2942 (71.1%)	2692 (70.0%)			1,870 (72.8%)	1,870 (72.8%)		
Tumor size			<0.001				1.0	
≤2 cm	203 (4.9%)	276 (7.2%)			121 (4.7%)	121 (4.7%)		
2–5 cm	717 (17.3%)	775 (7.2%)			480 (18.7%)	480 (18.7%)		
5–10 cm	933 (22.5%)	870 (22.6%)			578 (22.5%)	578 (22.5%)		
>10 cm	2,286 (55.2%)	1,923 (50.0%)			1,388 (54.1%)	1,388 (54.1%)		

Furthermore, 2,567 female and 2,567 male patients were perfect matched pairs for better analyses, baseline clinicopathological characteristics after perfect propensity-score-matched (PSM) analyses were also showed in Table 2, the differences between two groups were eliminated with regard to primary tumor site, age, therapy, marital status, race, grade, tumor size (all factors P = 1.0). After PSM, the median survival time of male patients was 106 months (95% CI [98.90–113.10]), while matched female patients were 162 months (95% CI [143.26–180.74]). Consistently, female patients had a significantly longer overall survival time than male patients in GIST (HR = 1.470, P < 0.0001, Fig. 2B).

Identification of biological processes and signaling pathways distinguishing certain subtype by gene set enrichment analysis (GSEA)

Significant gene sets were visualized as interaction networks with Cytoscape and Enrichment Map (Merico et al., 2010). For analysis of male and female expression differences, we chose 18 female vs. 18 male matched samples with equal tumor size for Cytoscape and Gene Set Enrichment Analysis. Compared to female patients, interferon signaling, cytokine signaling in immune system, immune checkpoint (PD1, TCR, CTLA4) and circadian expression were significantly up-regulated in subtype of male patients (Figs. 3A–3D). However, pathways of oxidative phosphorylation, TCA cycle and respiratory electron transport were up-regulated in subtype of female patients compared to male patients (Figs. 3A, 3E–3G).

Figure 3 Gene set enrichment analyzes between subtypes by gender and subtypes by lncRNAs.

(A) Cytoscape map showing the upregulated and downregulated biological pathway in 18 male patients compared to matched 18 female GIST patients. (B–D) GSEA maps showing main upregulated pathway in male patients. (E–G) GSEA maps showing main upregulated pathway in female patients. (H) Cytoscape map showing the upregulated biological pathway in the subtype G3/4 GIST patients (tumor size > 5 cm) compared to G1/2 GIST patients (tumor size < 5 cm). (I–K) GSEA map showing the main upregulated pathway in G1/2 patients. (L–N) GSEA map showing the main upregulated pathway in G3/4 patients. Orange nodes mean upregulated gene sets, blue nodes mean downregulated gene sets, node size is proportional to the total number of genes within each gene set.

Subtypes G1 and G2, G3 and G4 were combined for analysis, and referred as G1/2 (small size) and G3/4 (large size) respectively. Compared to subtype G1/2, most signaling pathways showed up-regulated in subtype G3/4, including DNA repair, carbohydrates metabolism, protein metabolism and lysosome pathway which might be involved in carcinogenesis or development of GIST (Figs. 3H, 3L–3N). A few signaling pathways such as olfactory signaling, olfactory transduction signaling and hedgehog pathway were up-regulated in subtype G1/2 compared to G3/4 (Figs. 3I–3K).

These analysis results suggested that different gender and lncRNAs subtypes of GISTs might arise from distinct abnormalities of biological signaling pathways.

lncRNA Markers selection and associated expression pattern between four subtypes

A total of 32 featured lncRNAs were identified to be significantly different expression between subtype G1/2 and subtype G3/4 by the Comparative Marker Selection algorithm, and their expression style were plotted in Fig. 4A. As lncRNAs’ functions are closely associated with their transcript abundance (Du et al., 2013), six lncRNAs LRRC75A-AS1, HYMAI, MALAT1, NEAT1, XIST and FTX with high expression value were recruited for further study.

Figure 4 LncRNAs markers selection.

(A) Comparative Marker Selection algorithm computed out 32 lncRNAs and their expression levels in four lncRNA clustering subtypes were plotted. (B) Functional enrichment analyses of the LRRC75A_AS1 co-expressed protein-coding genes by KEGG and GO analysis.

The expression of most featured lncRNAs were significantly downregulated in GIST progression subtype G3/G4 compared with subtype G1/G2, whereas LRRC75A-AS1 showed the opposite upregulation (Fig. 4A). High expression of LRRC75A-AS1 was positively associated with tumor diameters and seemed to be oncogene in GIST. XIST was specifically expressed in female GIST patients and was dramatically down-regulated in G3/G4 subtypes, which suggested XIST was significantly inhibited during tumor progression and it seemed to have a protective effect in female GIST patients. The expression of HYMAI, NEAT1, XIST and FTX were negatively correlated with tumor diameter and significantly down-regulated during GIST progression (Table 3). Overall, XIST down-regulation may suggest the malignant transformation for female GIST patients, while down-regulation of HYMAI, NEAT1 and FTX, or up-regulation of LRRC75A-AS1 may suggest the malignant transformation for both female and male patient, which may be the promising biomarkers irrespective of small intestinal or gastric origin.

Table 3 Co-relation analysis between featured lncRNA and clinical characteristics.

Pearson
correlation	LRRC75A-AS1	HYMAI	MALAT1	NEAT1	XIST	FTX	
Age	Coef.	0.059	0.163	−0.314	−0.424*	0.179	−0.244	
p	0.747	0.373	0.080	0.016	0.327	0.179	
Dimeter	Coef.	0.610**	−0.580**	0.171	−0.415**	−0.330**	−0.439**	
p	0.000	0.000	0.189	0.001	0.009	0.000	
Group_k4	Coef.	0.711**	−0.680**	0.196	−0.541**	−0.646**	−0.629**	
p	0.000	0.000	0.130	0.000	0.000	0.000	
Notes:

Coef.: Pearson correlation coefficient.

* P < 0.05.

** P < 0.01.

Protein-coding genes co-expressed with LRRC75A-AS1 and functional prediction

To further investigate the potential pathways associated with LRRC75A-AS1, the co-expressed protein-coding genes were computed using Pearson correlation coefficients (PCC) by R package LINC. There are 500 genes correlating with LRRC75A-AS1 (coefficient > 0.5). Gene ontology (GO) and Kyoto Encyclopedia of Genes and Genomes (KEGG) analysis showed that LRRC75A-AS1 co-expressed protein coding genes were mainly enriched in cadherin binding, unfolded protein and cell adhesion molecule binding, FK506 binding, isomerase and NADH activity (Fig. 4B). Taken together, these findings indicated that LRRC75A-AS1 may contribute to tumor malignancy by interacting with protein coding genes that involved in the above biological pathways.

lncRNA expression and CNA aberration

LRRC75A-AS1 as a novel candidate oncogene in chromosome region 17p11.2–p12 had been reported in human osteosarcoma to have a significant association between copy number and expression level (Both et al., 2012; Both et al., 2016). Although it also acted as an oncogene in GIST, LRRC75A-AS1 expression level showed no significant association with copy number alteration in our data analysis. Chromosome region 17p11.2–p12 showed a hotspot site in GIST, where LRRC75A-AS1 was significantly upregulated during tumor progression. Meanwhile chromosome region 17p11.2–p12 also contained protein-coding genes, RAI1, MSI2 and SMCR8, which were in co-expressed network of lncRNA HYMAI, NEAT1 and FTX (Fig. 5). Thus, chromosome region 17p11.2–p12 may contribute to the oncogenic procession in malignant GIST.

Figure 5 FiIntegrated circos association map of signature lncRNAs, co-expressed protein-coding genes and copy number.

The outer ring shows ideogram of a normal karyotype. The next outermost red and blue labels index represents the most relevant lncRNAs markers. Red represents lncRNA positive correlation with expression level changes and blue represents negative correlation. The inner ring represents copy number as a function of genomic coordinates. Red represents amplification and green represents deletion. In the center of the figure, the connector showed the co-expressed protein-coding genes of representative abundant lncRNAs (LRRC75A-AS1, HYMAI, MALAT1, NEAT1, XIST and FTX (P < 0.001)).

Discussion

Molecular expression profiles including the aberrant expressions of lncRNAs (Chen et al., 2014; Qi & Du, 2013), have provided more information to help identify the molecular subtypes, grades and prognosis of malignancy of various tumors (Chen et al., 2014; Sadanandam et al., 2013), as well as in GIST (Lee et al., 2016; Niinuma et al., 2012). In this study, we first used mRNA gene expression profiles to classify the GIST samples and did not show any clinical meaning. Instead, by profiling lncRNA expression, the GIST samples could be well clustered into four molecular subtypes with distinct biological processes and different phenotype in their clinical character. Thus, similar to other studies (Hessels et al., 2003; Prensner & Chinnaiyan, 2011), lncRNAs may have higher specificity than mRNAs, and be more suitable to serve as prognostic and/or predictive markers for GIST. In our present study, tumor location (stomach or intestine origin), tyrosine kinase mutation status and age did not show any association with lncRNA expression profile. The major finding in our present study was that lncRNA expression profile well clustered the GIST samples into small size (<5 mm) and large size tumors (>5 mm) which is a fundamental index for GIST malignancy.

Interestingly, the gender had strong influence on clustering by lncRNA expression profile in small size tumor (G1/G2). High expression of gender-related lncRNA, such as XIST, FTX and TTTY15 among the limited lncRNA-probes may attribute to this gender-oriented clustering. The great weight coefficient given by these gender-related lncRNAs were obvious in small size tumor (G1/G2 < 5 cm), and was rapidly lost when the tumor became malignant (G3/G4, > 5 cm) and the expression of XIST and FTX were significant inhibited. This expression pattern of XIST in GIST was totally different from other tumors in previous published studies (Ma et al., 2017) and meta-analysis (Zhu et al., 2018). Elevated lncRNA XIST expression predicts poor OS, poor disease free survival (DFS), larger tumor size, increased distant metastasis and advanced tumor stage in esophageal squamous cell carcinoma, pancreatic cancer, colorectal cancer, gastric cancer, nasopharyngeal carcinoma, non-small cell lung cancer (Zhu et al., 2018) and hepatocellular carcinoma (Ma et al., 2017). Here we demonstrated that female GISTs had a good OS and the gender-related lncRNAs XIST, FTX might play a protective role for GIST aggression.

The SEER data with 7983 GIST patients showed the favorable survival rate for female patients (P < 0.0001). In 2012, the lancet oncology published a pooled population-based cohorts consisting of 920 patients with GIST indicated that male sex were independent adverse prognostic factors (Joensuu et al., 2012), as well as large tumor size, high mitosis count, non-gastric location, presence of rupture. Here we further investigated the molecular mechanism underlay clinical differences between gender. GSEA results showed in male patient, interferon signaling, cytokine signaling in immune system, immune checkpoint and circadian expression was significantly up-regulated while tricarboxylic acid cycle (TCA) and respiratory electron transport pathways were down-regulated. Immune microenvironment profiling of gastrointestinal stromal tumors (GIST) shows gene expression patterns associated to immune checkpoint inhibitors response (Pantaleo et al., 2019). Gene expression profiles (GEP) and immunohistochemistry (IHC) from 31 KIT/PDGFRA-mutant GIST supported the presence of immune infiltrate, IFN-gamma-induced immune signature (EIIS) and the T-cell-inflamed signature (TIS), which suggests that GIST may benefit from immunotherapy along with tyrosine kinase inhibitors (Pantaleo et al., 2019). Abundant infiltrating immune cells were found in PDGFRA-mutant GISTs and PD-L1 expression was negatively associated with tumor size (Sun et al., 2020). GSEA between small tumor size (G1/G2) and large tumor size (G3/G4) also showed significant differences, and gene set involved in carcinogenesis were identified as following: DNA repair, carbohydrates metabolism, protein metabolism and lysosome pathway were up-regulated in subtype G3/G4. A few signaling pathways such as olfactory signaling, olfactory transduction signaling and hedgehog pathway were up-regulated in subtype G1/2. The previous results from immunohistochemical, immunofluorescence and ultrastructural techniques (Iruzubieta et al., 2020) showed Hedgehog signaling pathway activation (a pathway related with tumoral features such as proliferation, migration or stemness) mediated by primary cilia (an antenna-like structure based on microtubules presented in GIST cells) would be fundamental in tumoral microenvironment control of GIST cells for their maintenance, differentiation and proliferation. Hedgehog signaling could also activate KIT expression irrespective of mutation status, offering a novel approach to treat imatinib-resistant GIST (Tang et al., 2016). We also identified five lncRNAs with abundant expression (LRRC75A-AS1, HYMAI, NEAT1, XIST and FTX) were closely associated with tumor progression, which may suggest to be the biomarker for the malignancy of GIST. Particularly, LRRC75A-AS1 was positively corelated with tumor diameters and maybe an oncogene in GIST. Most recent study showed that Long non-coding RNA LRRC75A-AS1 facilitates triple negative breast cancer (TNBC) cell proliferation and invasion via functioning as a ceRNA network of LRRC75A-AS1/miR-380-3p/BAALC in accelerating TNBC development, indicating new promising targets for TNBC treatment (Li et al., 2020). Previous studies reported lncRNAs HOTAIR (Lee et al., 2016; Niinuma et al., 2012), H19 (Badalamenti et al., 2019) and AOC4P (Hu et al., 2018) are associated with the onset and progression of gastric GISTs and drive malignant character in GIST. In present study, the expression HOTAIR from the chip-derived data was very low, and both HOTAIR and H19 did not shown any association with GIST clinicopathological value. AOC4P was not covered by this chip platform of Affymetrix HG-U133 Plus 2.0 arrays, analysis cannot be performed.

Coincidentally, most of the featured lncRNAs are referred as chromatin associated RNAs (caRNAs) (Sridhar et al., 2017), including previously known XIST, NEAT1, MALAT1 (West et al., 2014). The chromatin associated long ncRNAs exert regulatory control on gene function through interaction with chromatin-associated proteins (Sridhar et al., 2017; West et al., 2014). So, modulation of chromatin structure maybe a mechanism of GIST carcinogenesis and aggression. We identified nine protein-coding genes, which were significantly co-expressed with these featured lncRNAs and also differentially expressed between tumor stage. One of these protein genes, MSI2 was well defined as the loci target of NEAT1 previously (Sridhar et al., 2017; West et al., 2014). Oncogene-like LRRC75A-AS1, protein-coding co-expressed with caRNAs genes RAI1, SMCR8, and previous known NEAT1 targeted MSI2, were all at 17p11.2–p12. Thus, chromosome region 17p11.2–p12 may contribute to the oncogenic procession in malignant GIST, which was similar to previous published study in human osteosarcoma (Both et al., 2012; Both et al., 2016). The circos plot showed that LRRC75A-AS1 had no significant association between copy number and gene expression level at site of 17p11.2–p12. Thus, expression activation of LRRC75A-AS1 in GIST was not similar as mechanism of concerted amplification-mediated overexpression with copy number aberration in human osteosarcoma (Both et al., 2012).

We also acknowledged some limitations of this study. First, the main drawback of present study is the lack of information on mitotic rate, which is one of two consensus factors for estimating the relative risk of GISTs (Fletcher et al., 2002). And the information of survival and clinical stage is not available either. Second, Affymetrix HG-U133 Plus 2.0 arrays did not cover all the possible lncRNAs present, the lncRNAs candidates identified here may not represent the complete lncRNA profiles underlying GIST clinical progression (Chen et al., 2014). To accurately and comprehensively elucidate the role of lncRNAs in classification of GIST, more comprehensive profiling studies and laboratory and clinical researches are needed.

Conclusions

In summary, we identified four molecular subtypes in GIST based on the lncRNA profiles. LncRNA expression between female and male showed distinct lncRNA profiles gene-set pathways, and female GIST patients also showed favorable prognosis. LRRC75A-AS1 (Gene ID: 125144, LRRC75A antisense RNA 1) may present an oncogene during the GIST carcinogenesis. Modulation of chromatin structure by caRNAs, such as XIST, NEAT1 and MALAT, maybe a mechanism of GIST carcinogenesis and aggression. Although the possible functions of many identified lncRNA genes need to be further investigated, our study of the lncRNA based classification may provide an efficient classification tool for marker selection and clinical progression evaluation of GIST.

Supplemental Information

Supplemental Information 1 Case summary from microarray data GSE8167 and GSE17743.

Click here for additional data file.

Supplemental Information 2 Raw data source.

The raw data source of gastrointestinal stromal tumors patients analysed in this article.

Click here for additional data file.

Additional Information and Declarations

Competing Interests

Author Contributions

Data Availability

The authors declare that they have no competing interests.

Yan Zhao performed the experiments, analyzed the data, prepared figures and/or tables, authored or reviewed drafts of the paper, and approved the final draft.

Xinxin Liu conceived and designed the experiments, performed the experiments, analyzed the data, prepared figures and/or tables, authored or reviewed drafts of the paper, and approved the final draft.

Keshuai Xiao performed the experiments, analyzed the data, prepared figures and/or tables, and approved the final draft.

Liwen Wang analyzed the data, prepared figures and/or tables, authored or reviewed drafts of the paper, and approved the final draft.

Yuping Li performed the experiments, prepared figures and/or tables, and approved the final draft.

Mingyun Kan performed the experiments, prepared figures and/or tables, and approved the final draft.

Zhiwei Jiang conceived and designed the experiments, analyzed the data, authored or reviewed drafts of the paper, and approved the final draft.

The following information was supplied regarding data availability:

The raw data is available in the Supplemental File and at NCBI GEO: GSE17743, GSE8167, GSE20709.

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
