# Peer review of "Clinicopathological value of long non-coding RNA profiles in gastrointestinal stromal tumor"

_PeerJ, doi:10.7717/peerj.11946_

## Round 0.1 · original submission · Major Revisions

I would appreciate it if the authors could improve the manuscript to address a few points raised by reviewers such as Hazard Ratio, laboratory evaluation, and pairing criteria.

·

Basic reporting

Zhao et al describe a data-mining approach using publicly available databases, for the identification of potential long-non coding RNAs (LncRNA) in the pathogenesis of Gastrointestinal Stromal Tumors (GIST). I find the concept quite interesting and I would support its publication if the authors address several major comments on this work.

1. The language throughout the manuscript needs significant improvement and editing. There are several occasions that the language limits the proper comprehension of the actual meaning for the reader. Some of the examples include lines 24, 28, 37, 44, 47, 58, 66, 70, 145-148, 189, 191-192, 232-241, 251.

2. The authors also need to improve and update their reference utilization. In the introduction section, they should provide a more recent and updated source for the clinical aspects of GIST tumors (line 57-60). In addition, in the same section, the authors should expand the scientific background on the role of lncRNA in GIST. More specifically, in line 72 they should include examples of lncRNA that affect prognosis. Furthermore, in the line 73-78, the authors have not included important references for the role of lncRNA in GIST for MALAT1, H19 ( https://doi.org/10.1155/2019/5458717, PLOS ONE 13(12): e0209342.), AOC4P (Onco Targets Ther. 2018; 11: 6259–6269., World J Meta-Anal. Jun 28, 2020; 8(3): 233-244) and extended reviews for the role of lncRNA in imatinib resistant GIST (Oncol Lett. 2019 Feb; 17(2): 2283–2295, https://doi.org/10.1590/1414-431x20198399). Also, the authors should edit their introduction in order to include further information regarding the available treatment approaches and molecular mechanisms that contribute to the pathogenesis of GIST. Another area of significant improvement could be the discussion section. The authors, in line 333-335, falsely claim the lack of knowledge on the molecular pathways affected by lncRNA LRRC75A-AS1. They cite several papers for its role in osteosarcoma, ignoring its major contribution in the pathogenesis on Triple Negative Breast Cancer (Cell Death Dis 11, 643 (2020), Int J Biol Sci. 2020; 16(2): 251–263. )

3. Another important aspect of improvement is the preparation of the figures. The figures are not consistently labeled, are very small to read and in many cases difficult to follow. I strongly suggest the authors to use high-quality images and prepare them using professional software (GraphPad Prism, Python, Adobe Illustrator)

Experimental design

Although the overall technical approach is correct, I have some concerns regarding the experimental design.


1. Although the original approach is unbiased and the question well defined, with the proper analysis of publicly available datasets for GIST patients, I strongly believe that the later focus only on LRRC75A-AS1 is not justified based on the overall results and the literature. I find the data in figure 5 of little importance and contribution to the field. The authors should revise their approach and generate data showing the pathways that their proposed lncRNAs candidates affect, in a comprehensive unbiased way. I strongly suggest that the authors should show the effect of the expression of these lncRNAs on survival and clinical stage, if this information is available.

Validity of the findings

1. In figure 2A and 2B, the author should include the Hazard Ratio in their analysis and figures, and calculate the logp-rank for their suggested survival curves.

2. Last but not least, the authors should provide the pairing criteria used for the male and female patients, since this can lead to significant confounding bias. Furthermore, the authors should provide the list of matched patients with statistical proof that their matching is statistically correct. After that, the authors should revise their model and figures.

Additional comments

Zhao et al provide an interesting concept in the already established role of lncRNAs in GIST patients. The authors are strongly advised to re-evaluate several aspects of their manuscript and I would be positive on its publication.

Reviewer 2 ·

Basic reporting

The description and explanation of this manuscript are clear and nicely readable. All the figures and tables are well organized, except missing a note for * and ** in Table 3. It looks that table 2 is informative, but you need to explain more of any important information we can get from it.

Experimental design

Do you have the gene set enrichment analysis in a subtype of female patients? Any difference between male and female? Can you explain more about why XIST down-regulation may suggest the malignant transformation?

In the ideogram of figure 5, why didn’t you put Chr Y for the whole genome co-expressed protein-coding genes and copy number variation analysis?

Validity of the findings

In the review version, the resolution of figure 1B is not clear to figure out each vertical annotation that is associated with each subtype, but the expression pattern of these 2448 lncRNAs in G1 and G4 subgroups looks similar, and as well as G2 and G3. How can you explain this situation?

Additional comments

This manuscript described a process of how to start by profiling lncRNA expression to detect a specific lncRNA as a prognostic or predictive marker for GIST, which may have higher specificity than using mRNAs. By using different algorithms, LRRC75A-AS1 was detected that showing a higher expression level in the malignant GIST tumor (G3/G4 subtypes). This result could be used for the laboratory evaluation and verified if it can be used as a prognostic marker or progression diagnostic for GIST.

---

## Round 0.2 · Minor Revisions

Please address those concerns from Reviewer 2.

·

Basic reporting

Zhao et al have provided an extensively revised manuscript that strongly support their model. They have improved their figure quality tremendously making it very easy to follow.

Experimental design

The new data added in the study are well controlled and explained.

Validity of the findings

No comments.

Additional comments

The authors returned a significantly improved version of their work that meets the advanced scientific criteria of the journal. I support the publication of this concept.

Reviewer 2 ·

Basic reporting

The authors addressed the concerns and questions properly and accordingly, the revised version of the manuscript seems to be well written and should provide useful results. However, I still have some concerns and suggestions:
There are 9 images in Fig.1, but in your manuscript, you haven’t explained or mentioned each of the figures you listed. Moreover, in line 186 and line 192, your descriptions are not corresponding to the figures you mentioned. You need to revise those contents and add more explanation corresponding to all the images in figure 1. Similarly, you haven’t described all the figures in figure 3.

Experimental design

No comment

Validity of the findings

No comment

Additional comments

No comment

---

## Round 0.3 · Minor Revisions

You have not fully addressed the questions raised by reviewer 2 in the previous revision. Please respond to them carefully by adding more details about Figures 1 and 3.

---

## Round 0.4 · Minor Revisions

I would suggest the authors add more explanation about your figure 1 and 3, not just rephrase the sentence without adding any new discussion and interpretation.

---

## Round 0.5 · accepted · Accept

Congratulations - your manuscript is accepted for publication.